# Cometabolic Biodegradation of Hydrazine by *Chlorella vulgaris*–Bacillus Extremophilic Consortia: Synergistic Potential for Space and Industry

**DOI:** 10.3390/life15081197

**Published:** 2025-07-28

**Authors:** Yael Kinel-Tahan, Reut Sorek-Abramovich, Rivka Alexander-Shani, Irit Shoval, Hagit Hauschner, Chen Corsia, Ariel Z. Kedar, Igor Derzy, Itsik Sapir, Yitzhak Mastai, Ashraf Al Ashhab, Yaron Yehoshua

**Affiliations:** 1The Algal Biotechnology Center, The Mina and Everard Goodman Faculty of Life Sciences, Bar-Ilan University, Ramat Gan 5290002, Israel; yaelkinel@gmail.com (Y.K.-T.); chenkorsia@gmail.com (C.C.); arielkedar@gmail.com (A.Z.K.); 2The Dead Sea & Arava Science Center (DSASC), Masada National Park, Mount Masada, Dead Sea, Masada 8691000, Israel; reut.sorek@gmail.com (R.S.-A.); rivkasa@adssc.org (R.A.-S.); ashraf@adssc.org (A.A.A.); 3The Kanbar Core Facility Unit, The Mina and Everard Goodman Faculty of Life Sciences, Bar-Ilan University, Ramat Gan 5290002, Israel; irit.shoval@biu.ac.il (I.S.); hagit.hauschner@biu.ac.il (H.H.); 4VTS Energy Ltd., Petah Tikva 4951939, Israel; igor@vts-energy.com; 5School of Mechanical Engineering, Afeka Tel Aviv Academic College of Engineering, Tel-Aviv 6910717, Israel; itsiks@afeka.ac.il; 6Department of Chemistry and Institute for Nanotechnology and Advanced Materials, Faculty of Exact Sciences, Bar-Ilan University, Ramat Gan 5290002, Israel; mastai@biu.ac.il; 7Eilat Campus, Ben-Gurion University of the Negev, Beer-Sheva 84105, Israel

**Keywords:** hydrazine, *Chlorella vulgaris*, extremophile, *Bacillus*

## Abstract

Hydrazine, a highly toxic and reactive compound widely used as rocket fuel, poses significant environmental and health risks, particularly in long-term space missions. This study investigates the cometabolic capacity of *Chlorella vulgaris* and seven extremophilic *Bacillus* spp. strains—isolated from the arid Dead Sea region—to tolerate and degrade hydrazine at concentrations up to 25 ppm. The microalga *C. vulgaris* reduced hydrazine levels by 81% within 24 h at 20 ppm, while the *Bacillus* isolates achieved an average reduction of 45% over 120 h. Identified strains included *B. licheniformis*, *B. cereus*, and *B. atrophaeus*. Co-culture experiments demonstrated that *C. vulgaris* and *B. cereus* (isolate ISO-36) stably coexisted without antagonistic effects, suggesting a synergistic detoxification interaction. Flow cytometry revealed that most bacteria transitioned into spores under stress, highlighting a survival adaptation. Titanium, representing a biocompatible material common in aerospace hardware, did not inhibit microbial growth or hydrazine degradation. These findings underscore the potential of Dead Sea-derived microbial consortia for cometabolic hydrazine detoxification and support the feasibility of converting spacecraft components into functional photobioreactors. This approach offers dual-use benefits for space missions and industrial wastewater treatment. Future studies should investigate degradation pathways, stress resilience, and bioreactor scale-up.

## 1. Introduction

Hydrazine (N_2_H_4_), a highly toxic and reactive component of rocket fuel, presents a major challenge in space exploration and industrial wastewater treatment. Its presence in closed-loop systems can jeopardize biological life-support modules, and its detoxification is essential to ensure sustainable long-term space missions and environmental safety. One potential solution involves biodegradation of hydrazine using extremophilic microbial consortia, capable of transforming it into less harmful byproducts under controlled conditions.

Converting spacecraft fuel tanks into photobioreactors for cultivating microalgae and bacteria is a promising solution for various life support systems. Regenerative food production systems are in demand due to recent advances and cost reductions in space travel and habitat construction [1,2,3]. Due to their high biomass-to-nutrient ratio, microalgae have long been suggested as an alternative high-value food source [4,5,6]. *Chlorella vulgaris* is recognized as a well-researched candidate for food and other beneficial applications based on its proteins, lipids, carbohydrates, pigments, minerals, antioxidants, and vitamins [7]. Reutilization of out-of-commission spacecraft parts, such as fuel tanks and excess fuel, which are no longer needed once the spacecraft reaches its destination, can significantly reduce costs by decreasing the overall mission mass.

One potential challenge is the toxicity of the rocket fuel hydrazine to algae, bacteria, and other organisms [8]. Previous studies showed that concentrations of 1.03 ppm and 0.8 ppb caused 50% growth inhibition in microalgae *Selenastrium capricornutum* and *Dunaliella tertiolecta* spp., respectively [9,10]. The concentration range of hydrazine toxicity to algae depends on the species and growth conditions. For example, the half-maximal effective concentration (EC50), which reduces growth rate by 50 percent, for the freshwater green microalga *Selenastrum capricornutum*, ranges from 0.4 to 30 ppb [10]. The safe concentration (SC) range of hydrazine to algae is 0.1 to 1 ppb under oligotrophic conditions [11]. These data suggested analyzing the response of *C. vulgaris* to different hydrazine concentrations, starting at 0.5 ppb.

Beyond hydrazine itself, related compounds such as unsymmetrical dimethylhydrazine (UDMH) and monomethylhydrazine (MMH)—commonly used in aerospace propulsion and military-grade fuels—are even more toxic and environmentally persistent. These substances are volatile, carcinogenic, and pose considerable challenges for conventional remediation. Therefore, developing robust biodegradation systems that can address both hydrazine and its derivatives is a crucial environmental and biotechnological priority.

Microbial biodegradation can sometimes be considered a preliminary step in the decontamination of toxic organic pollutants such as hydrazine. Extremophile bacteria have long been recognized as promising candidates for bioremediation and biodegradation of fuels and other hazardous compounds [12,13,14,15]. For instance, *Pseudomonas* spp. can utilize hydrazine as a nitrogen source [16], while Nitrosomonas, an ammonia-oxidizing proteobacterium, can tolerate concentrations of up to 95 ppm and degrade hydrazine primarily to N_2_ [17].

The Dead Sea region, where our bacterial isolates originate, represents one of the most extreme ecosystems on Earth, with hypersalinity (~34% total dissolved salts), intense solar radiation, low oxygen levels, and frequent temperature fluctuations. These environmental pressures select for microbes with exceptional stress tolerance and metabolic plasticity. The *Bacillus* strains used in this study were selected from a desert bacterial library established at Al-Ashhab Lab, constructed from environmental samples collected across the Dead Sea region, including soils, rhizospheres, and plant phyllospheres [18]. These extremophilic isolates have previously demonstrated key plant-growth-promoting traits, such as atmospheric nitrogen fixation [19], phosphorus solubilization [20], and siderophore production [21], which may also contribute to their cooperative potential in microbial consortia.

Since the bacteria and microalgae are to coexist in the reactor system, the interaction of microbial species with the algae of choice must be carefully examined, as heterotrophic bacteria within algae cultures can impact the health, nutrient consumption, and growth of the cultures [22,23,24]. Specifically, *Chlorella vulgaris* is inhibited when the bacterial concentration is high due to resource competition, while at low concentration it has a positive effect on algae growth [25]. Examination of symbiotic relationships of *C. vulgaris* cultures with three bacterial species (*Pseudomonas alcaligenes*, *Elizabethkingia miricola,* and *Methylobacterium radiotolerans*) showed enhancement with *P. alcaligenes*, and a negative effect with *E. miricola* and *M. radiotolerans* [26]. Kim et al. reported that different nitrogen sources and concentrations had a positive effect on the symbiotic interactions with *Microbacterium kitamiense* [27]. *Bacillus* spp. possess antifungal and antibacterial properties, and can also inhibit the growth of some algae [28,29]. *B. pumilus* bacteria, isolated from cultures of marine microalga *Nannochloropsis salina* and seaweed *Padina pavonica*, inhibited algal growth [30,31].

Here, we examine the co-culture and hydrazine tolerance of *C. vulgaris* and extremophilic *Bacillus* spp. at different concentrations of hydrazine hydrate, to assess their potential for cometabolic biodegradation. The objective of this study is to evaluate the hydrazine tolerance and degradation capacity of these organisms—individually and in co-culture—under conditions relevant to closed-loop systems such as space habitats and hydrazine-contaminated industrial environments. By exploring their cometabolic interactions, we aim to identify synergistic microbial strategies for sustainable detoxification.

## 2. Materials and Methods

### 2.1. Bacterial Growth with Hydrazine

Sixty-two bacterial isolates [18] were screened for hydrazine tolerance: after overnight (ON) incubation in 4 mL of LB broth (HiMedia Laboratories, Thane, India) at 37 °C with continuous shaking at 150 rpm, 0.5 mL were transferred to freshly made 3.5 mL LB, and the optical density (OD) at 600 nm was checked after a two-hour incubation period. Once the bacterial isolates were confirmed to be in mid-log phase, 100 µL of bacterial suspension were dispensed into a 96-well plate (Corning^®^ 96-well clear flat bottom polystyrene untreated microplate) and divided into three main sections: (i) bacterial growth without hydrazine (100 µL of bacterial suspension + 100 µL of double-distilled water (DDW)); (ii) bacterial growth with hydrazine (25 ppm, equivalent volume); (iii) controls (LB, LB + DDW, hydrazine + LB, and hydrazine + DDW). A solution of hydrazine hydrate puriss. p.a. (Sigma-Aldrich, Jerusalem, Israel), 24–26% in H_2_O at room temperature (RT) was prepared at an initial concentration of 50 ppm. The 96-well plates were incubated for 24 h at 25 °C inside a spectrophotometer (BioTek Synergy HTX, Agilent, Petach Tikva, Israel), and growth kinetics were measured every 20 min at 600 nm. Data analysis was based on the average OD of three replicates compared to a blank reading (*n* = 3).

### 2.2. Identification of Extremophile Bacterial Isolates

Seven selected isolates were identified by 16S rRNA gene sequencing: DNA was extracted using the Dneasy blood and tissue kit (Qiagen, Valencia, CA, USA) according to the manufacturer’s instructions, and PCR amplification was carried out using two 16S sets of universal primers: 27F-907R and 341F-1492R. Amplification of the gyrA and GroEL gene coding regions was conducted to obtain a more precise taxonomic definition within the *Bacillus* spp. group, following the protocol of [32]. The obtained sequences were edited and aligned using Geneious Prime^®^ (V. 2021.2). Multiple sequence alignment was performed using the MUSCLE software version 3.8.425 [33], and a neighbor-joining phylogenetic tree was constructed using the Tamura-Nei algorithm [34].

### 2.3. Algae Growth and Hydrazine Tolerance

Cultivation conditions. *C. vulgaris*, a freshwater green microalga (strain number 211-11b from the SAG culture collection, University of Göttingen, Germany), was grown in Bristol medium [35] (Sigma-Aldrich, Israel) and kept under constant LED illumination: 70 µmol quanta/m^2^/s at 24 °C. The alga was grown in a 2 L Erlenmeyer flask with continuous aeration. Before each experiment, the culture was homogeneously distributed, and 20 mL was transferred to a series of 50 mL sterile Falcon tubes.

Hydrazine tolerance of *C. vulgaris.* The tolerance was examined in two phases: tolerance-1 (HT-1) and tolerance-2 (HT-2). HT-1: The series of experiments with different hydrazine concentrations, from 0.5 ppb to 20 ppm, is detailed in the Appendix A. HT-2: Cultures from HT-1 (1 ppm to 20 ppm hydrazine, Appendix A) were harvested by centrifugation (5000 rpm, 7 min) and reintroduced into fresh Bristol x2 media (OD = 1.8). Samples of new *C. vulgaris* cultures were also added to HT-2 as controls with different initial densities (OD = 1.8, 3.6 and 0.8 for FA, FAx2 and FA/2, respectively) as detailed in Appendix A. Hydrazine hydrate solution puriss. p.a., 24–26% in H_2_O (RT) was prepared at an initial concentration of 50 ppm in DDW. Hydrazine hydrate (aqueous) was added to cultures at a final concentration of 0.5 ppb to 20 ppm. Negative controls did not contain hydrazine.

The Falcon tubes were loosely sealed with a lid and placed on a horizontal shaker (Stuart, model SSL1; Hanoi, Vietnam) at 145 rpm in a chemical hood at a constant LED illumination (see above). The OD was measured at 680 nm using a Synergy H1 spectrophotometer (Agilent Technologies, Santa Clara, CA, USA). Algal growth rate (μ) was calculated as described [36].

### 2.4. Analysis of Hydrazine Concentration

Hydrazine concentration in bacterial and algal cultures was quantified using a colorimetric assay based on its reaction with para-dimethylaminobenzaldehyde (pDMAB), adapted from [37]. Under acidic conditions, hydrazine reacts with pDMAB to form a yellow chromophore (p-dimethylaminobenzalazine, D2N), which has a maximum absorbance at 454 nm. The intensity of absorbance at this wavelength is directly proportional to the hydrazine concentration and was quantified using a standard curve (see Appendix A).

For bacterial cultures (ISO-1, ISO-4, ISO-5, ISO-7, ISO-28, ISO-33, and ISO-36; isolate codes used for internal reference), isolates were grown overnight in LB medium, then subcultured into fresh LB and incubated for up to 3 h at 30 °C with shaking, until reaching OD_600_ ≈ 0.2. This ensured cells were in an early exponential phase prior to hydrazine exposure. A final concentration of 25 ppm hydrazine was then added to 20 mL cultures in 50 mL sterile Falcon tubes.

At specific time points (T0, T3, T6, T9, and T24 for bacteria; T0, T3, T6, T24, and T30 for algae), 1.5 mL of culture was withdrawn and filtered through a sterile 0.22 µm PES syringe filter (Merck Millipore, Burlington, MA, USA) to remove all cells. The cell-free supernatant was then mixed 1:1 with a 1:50 dilution of pDMAB (prepared in acidic ethanol). The reaction mixture was incubated at room temperature for 15 min to allow full chromophore development.

Following incubation, 100 µL of the reaction mixture was transferred to a 96-well clear flat-bottom microplate, and absorbance was measured at 454 nm using a microplate spectrophotometer. All measurements were performed in triplicate. Hydrazine concentrations were determined by interpolation from a standard curve generated using known hydrazine concentrations processed under identical conditions.

Cell growth was monitored independently of hydrazine quantification. For bacteria, OD_600_ measurements were taken from 100 µL unfiltered culture aliquots. For Chlorella vulgaris, algal growth was tracked by measuring OD at 680 nm, which corresponds to chlorophyll absorption and is standard practice for microalgae.

### 2.5. Co-Culture of Extremophiles and C. vulgaris

#### 2.5.1. Assessment of Bacterial Symbiosis

*C. vulgaris* cultures were assessed for symbiotic relations with bacterial isolates on agar plates. The following experiments were conducted. Three-week-old algal cultures (150 µL) were plated on fresh Bristol agar (1.5%) plates and incubated for three days under constant illumination at 25 °C (24 h light, 6000 K LED illumination). Each plate was divided into six sections, and in each section, a bacterial isolate was placed by immersing 6 mm diameter Whatman assay disks in bacterial cultures and placing them on the plate. Plates were incubated under the same conditions for one week, checked daily, and photographed at the end of the incubation period (see Appendix A).

#### 2.5.2. Growth Dynamics in Broth Media

To assess the growth dynamics of *C. vulgaris* with bacteria in broth media, the following experiment was carried out (Appendix A). Bacterial isolate ISO-36 (100 µL), grown in LB broth at log phase (0.8 OD) after ON growth at 37 °C, was added to a final volume of 4 mL. Bacteria were entered as a 1:10 volume compared to the alga *C. vulgaris* (0.9 OD). Bristol 2× 80%, LB 20% 4 mL were distributed into 15 mL tubes. One ml was distributed into 24-well plates in triplicate, as summarized in Appendix A. Growth measurements were taken over 16 days, a total of 29 time points (0–367 h). Approximately every two days, a new test group was prepared for flow cytometry analysis.

#### 2.5.3. Flow Cytometry Analysis of Sub-Populations

Flow cytometry analysis was used to identify bacteria and algae based on the differences in their fluorescence intensities.

Flow cytometry analysis was carried out by a BD LSR Fortessa instrument (BD Biosciences, San Jose, CA, USA). Data was collected from the following channels: on the 488 nm laser—forward scatter (FSC) and side scatter (SSC), reflecting the size and granularity of the cells, respectively, and on the 640 nm laser—670/14 nm detector reflecting the auto-fluorescence of algae cells. The threshold was set at the minimum FSC. Bacteria appear as blue dots, but do not have fluorescence in the red spectrum (Appendix A). 

Cells were analyzed by imaging flow cytometry (ImageStream^®X^ Mark II imaging flow cytometer; Amnis Corp., part of EMD Millipore, Seattle, WA, USA). A mixed culture of bacteria and algae was analyzed (Appendix A). A 60× magnification was used for all samples. For each sample, 3000–5000 cells were collected, and data were analyzed using dedicated image analysis software (IDEAS 6.2; Amnis Corp., Seattle, WA, USA). Due to their chlorophyll, the algae have a pigment autofluorescence in the red spectrum (seen in ImageStream: channel 5–561 nm ex, 642–745 nm em. The bacteria, on the other hand, presented high side scatter intensity (ImageStream: dark field, SSC channel 6 (Appendix A) Some non-vital cells or broken particles, residing between the gated populations, were excluded from further analysis (Appendix A).

### 2.6. Growth with Titanium Plates

As space vessels and rocket fuel tanks are made of titanium alloy [38,39], we examined whether titanium affects *C. vulgaris* and bacterial growth. Five 50 mm titanium plates (5 cm^2^ surface area) were placed in algae test tubes with (i) 5 ppm and (ii) 20 ppm hydrazine hydrate and (iii) control without hydrazine (Appendix A, *n* = 3). Bacterial testing included titanium insertion with (i) 5 mL LB only without hydrazine, and (ii) control without Ti. Duplicates of bacterial isolates, as detailed in Section 2.3 (ISO-1, 4, 5, 7, 28, 33, 36), were incubated for 24 h at 37 °C with continual shaking at 150 rpm. This setup was included to confirm that titanium, as a biocompatible aerospace-grade material, does not interfere with microbial viability or hydrazine degradation, thereby supporting its potential use as a structural surface in integrated space bioreactors.

## 3. Results

### 3.1. Growth of Bacterial Isolates with Hydrazine

Sixty-two isolates from the DSASC extremophiles culture collection [18] were screened for their ability to grow in the presence of 25 ppm of hydrazine. There were three main growth patterns (Appendix A): (i) inhibition of isolates ISO-8, 9, and 19; (ii) enhanced growth of ISO-1, 4, 5, 7, and 33; (iii) further growth of ISO-28 and 36 compared to the control using hydrazine as an additional food source. The seven successful isolates were tested for 120 h, and hydrazine concentrations were measured during their growth (see Section 2.1). Our results indicate distinctly lower concentrations after 120 h: on average, there was a 45% reduction with bacteria, compared to a 7% reduction in the controls (Figure 1). For the first six hours, hydrazine concentrations were not significantly lower, but at 24 h, there was an average reduction of 9%. ISO-4 and 36 achieved a reduction of 15% and 51%, respectively.

While the hydrazine concentration in the LB-only control declined only modestly (by approximately 8–10% over 120 h), this suggests a minor abiotic effect, potentially due to slow chemical degradation, medium interactions, or limited evaporation. Nonetheless, the much greater hydrazine reduction observed in bacterial cultures (up to 51%) strongly supports a biologically driven degradation process. Therefore, while abiotic factors may play a small role, the dominant mechanism of hydrazine depletion in our cultures is attributed to microbial activity (Figure 1).

Based on growth measurements, the log phase for all bacterial cultures ranged from 3 to 24 h. Between 24 and 48 h, the growth was slower, but the population continued to increase. After 48 h, the stationary phase began, leading to a decline in the population (as evident in the 600 nm OD, see Figure 1, T_120_). At 48 h, the highest OD was obtained with ISO-36 (0.56), and the lowest with ISO-4 and 5 (both 0.23), indicating that ISO-36 was the least affected, reducing only 15% after 120 h. Also, after 120 h, ISO-4 had the lowest concentration of hydrazine (50%, Figure 1).

### 3.2. Identification of Extremophilic Bacterial Isolates

Following 16S rRNA sequencing, isolates were found to be close to various *Bacillus* spp. ISO-1 belongs to a branch that includes *B. licheniformis* (NR_118996), *B. piscis*, *B. paralicheniformis*, and *B. haynesii* (see Figure 2). ISO-5 and 7 formed a distinct cluster, with their closest relative being *B. paralicheniformis.* ISO-4 clustered with another strain of *B. licheniformis* (NR_116023) and ISO-36 clustered with strains of *B. cereus*, while ISO-28 clustered next to *B. atrophaeus*. The closest match to ISO-33 was *B. safensis*.

### 3.3. Growth of Algae with Hydrazine

A series of algal growth experiments with 50-1000 ppb different hydrazine concentrations was conducted showing no significant changes in the growth rate (see Appendix A and Appendix A).

During the 18-day period, the growth rates of low and high initial culture densities (0.091 ± 0.002 and 0.83 ± 0.01 OD, respectively) were examined. In the presence of 1 ppm hydrazine, algal-specific growth rate (μ) was 0.148 and 0.043 per day, only 3% and 4% higher than the respective control. At a high initial density, with 5, 10, and 20 ppm, µ was 0.056, 0.029, and 0.037, respectively, 11%, 28% and 8% lower than the corresponding control (Figure 3).

Higher concentrations of hydrazine, 10 and 20 ppm, partially inhibited algae growth during the lag phase. The log phase started after four days, compared to two days in the control group. At the end of the log phase, on day 14, the control group showed an OD of 1.50 ± 0.04, compared to 1.27 ± 0.05 and 1.41 ± 0.13 for the 10 and 20 ppm treatments.

We wanted to see if the lag phase would be shorter or the growth rate would increase as a result of potential adaptation in algal cultures previously exposed to hydrazine. Cultures exposed to 1 to 20 ppm hydrazine were transferred to a fresh medium with 20 ppm hydrazine (Figure 4; Appendix A). Controls were a “fresh” culture (FA) that was not exposed to hydrazine and Bristol medium with 20 ppm hydrazine and no algae (BR). We noticed the following changes in growth rate in previously exposed cultures (Figure 4). With respect to the control, the rates of A1, A5, and A10 were lower, by 75%, 1% and 31%, respectively, while that of FA was 27% higher. Except for A20, all other cultures exhibited growth patterns similar to the control group. A20 showed a decline, and on day 11, the OD was 50% lower than the initial value. After 24 h, all algae cultures had low hydrazine concentrations and exhibited lag phase growth characteristics until the fifth day (Figure 4).

Regarding hydrazine, we observed no major difference in degradation between fresh algae culture and those that were pre-exposed to hydrazine. Hydrazine concentration did not change in the control (Bristol medium w/o algae), indicating that hydrazine evaporation was not the main factor in the degradation. Previously, we observed that hydrazine levels decreased proportionally to the initial culture density, with higher reduction rates associated with relatively high culture density (Appendix A).

A1 and fresh algae (FA) behaved very similarly when exposed to 20 ppm of hydrazine. The concentration in the FA culture decreased by 52%, 70% and 85% at T_3_, T_6,_ and T_24_, respectively. These results are consistent with those of the other cultures (Figure 4); the average hydrazine concentration decreased by 42.8% at T_3_, 64.8% at T_6_, and 81.0% at T_24_.

### 3.4. Co-Culture of Bacillus spp. Isolates and C. vulgaris

Bacterial isolates ISO-7, 33, and 4 were selected for co-culture as they represent two different responses to hydrazine reduction, and also originate from two separate phylogenetic clusters. We wanted to test a representative from each group against *C. vulgaris* (without hydrazine first). These isolates, together with *C. vulgaris,* were spread on Bristol agar plates (see Section 2.5) and checked after one week of incubation (Appendix A). Bacterial growth was limited or did not occur at all on the Bristol plates, with no new bacterial colonies forming. This could be due to a lack of nutrients or perhaps due to the temperature being too low for optimal bacterial growth. After seven days of incubation, *C. vulgaris* was able to overgrow on the bacterial assay disks and cover the plate completely, as expected due to its autotrophic capabilities. Following this experiment, it was assumed that, on solid media, there would be no inhibitory or toxic compounds produced by the bacterial isolates that would harm or interfere with *C. vulgaris*. We continued with liquid media and also analyzed bacterial growth on mixed growth media (80% Bristol and 20% LB, see Appendix A). ISO-36 bacterial growth was the least affected by the hydrazine in the bacterial-hydrazine growth experiment (Section 3.2). We thus proceeded to test it with *C. vulgaris.*

Bacterial-algal growth dynamics without hydrazine suggested that *C. vulgaris* growth and chlorophyll benefited from the addition of LB medium, compared to Bristol medium only (Figure 5). The lowest OD measurements were of *C. vulgaris,* which exhibited an exclusively autotrophic growth pattern. The highest OD values were measured at day 10 for *C. vulgaris* in Bristol and LB (without ISO-36) and ISO-36 (*bacillus* spp.) on day 14. This indicated that mixed autotrophic and heterotrophic conditions enabled *C. vulgaris* to flourish earlier than bacteria growing exclusively on LB.

In the mixed-growth medium, the *C. vulgaris* specific growth rate was higher (μ = 0.54), compared to Bristol medium only (0.27) during the first seven days. For the same time period, the rate of ISO-36 *Bacillus* spp. in mixed medium was 0.77, compared to 0.81 in LB. *C. vulgaris* and *Bacillus* spp. co-culture had μ = 0.46.

The next challenge was to separate the bacterial population into its two possible types: vegetative and spores. This was achieved by analyzing the differences in appearance—vegetative bacteria are elongated, while spores are perfectly round and compact (Appendix A)—using imaging flow cytometry. The following sequence was performed. First, only focused bacterial cells were gated to exclude out-of-focus objects, achieved by plotting the gradient RMS of the bright field (BF) channel. The bacteria were then gated based on their size (area feature) to exclude large aggregates. Following this, we created a spot count feature to include only images containing single bacteria. Finally, the two populations were separated according to SSC intensity and the BF aspect ratio feature (the ratio between the minimum and maximum diameters). Vegetative bacteria were identified as having a low aspect ratio due to their elongated shape and mid-to-low levels of SSC intensity, while the round spores were identified as having a high aspect ratio and mid-to-high SSC intensity. For each treatment, the number of spores vs. vegetative bacteria was quantified. At the end of the experiment (day 16), most bacteria were spores, with a small number of vegetative ones (Appendix A).

### 3.5. Impact of Titanium on Growth Patterns

Results indicate that titanium, as tested in our experiments, did not have an adverse effect on algal or bacterial growth, with or without hydrazine. Average total OD values (at 600 or 680 nm) were similar, indicating that Ti had no significant effect on growth (Appendix A). Hydrazine concentrations also did not change due to the presence of titanium. Titanium containers were included to empirically confirm that microbial growth and hydrazine degradation remain unaffected when cultured in contact with spacecraft-relevant biocompatible materials. This validation supports the feasibility of using repurposed space hardware (e.g., fuel tanks) as microbial bioreactors in future space applications.

## 4. Discussion

### 4.1. Growth and Reduction in Hydrazine Hydrate

Hydrazine fuels are often used as rocket and jet propellants in both aircraft and spacecraft. Published toxicity studies were conducted to determine their impact on plants, animals, humans, algae, and a few microorganisms, with a primary focus on understanding mutagenesis by hydrazine [40].

Previous studies examined mixed cultures of bacteria to assess the bacterial metabolism of hydrazine fuels and their toxic effects. *Nitrosomonas* and *Nitrobacter* (nitrifying bacteria) were found to degrade up to 50% of hydrazine after seven days, and *Nitrosomonas* was found to degrade the hydrazine and release nitrogen gas as the primary metabolic product [17]. Other bacteria tested with hydrazine include *Enterobacter cloacea*, a soil heterotroph [41]; non-specific ammonium-oxidizing bacteria (AOB) and anammox bacteria [42]; *Achromobacter* sp., *Rhodococcus* B30, and *Rhodococcus* J10 [43].

Interestingly, the performance of singular ISO-36 in a mixed medium of Bristol and LB (80/20 ratio) was the same as that of ISO-36 with *C. vulgaris* in the same medium, indicating that the autotrophic *C. vulgaris* does not contribute to or inhibit the growth of ISO-36. However, as 80% of the medium was Bristol 2×, the growth of heterotrophic *Bacillus* spp. ISO-36 was sub-optimal, compared to LB pure media, as can be seen in the Results (Figure 5). In summary, experiments on growth in both agar and broth suggest that the *bacillus* spp. isolates can grow with *C. vulgaris*.

### 4.2. Effects of Hydrazine on Algae and Photosynthesis

In our experiments, the growth rate of *C. vulgaris* was influenced by hydrazine. Concentrations of 5 to 20 ppm resulted in slower algae-specific growth; however, at 1 ppm, the growth increased. When exposed to 20 ppm, previously exposed algae cultures exhibited lower growth rates. Specifically, the growth of A1, A5, and A10 cultures was slower than the control (w/o hydrazine).

Hydrazine is used in water treatment to prevent infrastructure corrosion caused by oxygen, as it removes oxygen via the reaction N_2_H_4_ + O_2_ → N_2_ + 2H_2_O [44]. Algal photosynthesis produces dissolved oxygen in the water medium, which hydrazine can remove. In previous studies, 20 ppm hydrazine was reduced by 50% at 25 °C within 90–150 min due to oxygen scavenging [45]. After 90 min, 1 ppm hydrazine was entirely eradicated (100%) and completely removed from the medium in the presence of dissolved O_2_ [45].

Accordingly, we observed that hydrazine levels changed in proportion to culture density. High density led to rapid hydrazine reduction; this may correlate with the oxygen scavenging mechanism, as found by Gaunt and Wetton [45] (Appendix A). Cultures previously exposed to hydrazine did not exhibit changes in the reduction rate (HT-2, Figure 4). Both observations suggest a physical, rather than biological, mechanism for hydrazine reduction in media.

In our experiments, *C. vulgaris* reduced hydrazine levels at concentrations of 5 to 20 ppm. It is possible that the remaining hydrazine infiltrated the algae cells [46,47,48]. Within the cells, it is likely that hydrazine acted on photosystem II, inhibiting photosynthesis [49,50]. Previous studies with *Synechococcus elongatus* (cyanobacteria) showed that O_2_ evolution and CO_2_ fixation were inhibited by hydrazine at 32–64 ppm [46] and confirmed that hydrazine also acted on photosystem II in *Chlorella* sp. ([46] and associated references). Another study showed that hydrazine and hydrazide derivatives inhibited *C. vulgaris* growth, possibly through interaction with aromatic amino acids in enzymes related to the biosynthesis of chlorophyll [48]. High concentrations of hydrazine are thus toxic and can cause algae cell death.

A further decrease in hydrazine concentration may be attributed to symbiosis with additional bacteria in the native algal culture. It has been shown that a cyanobacterial symbiont can use hydrazine in the process of nitrogen fixation in algal cultures [51]. Specifically, *C. vulgaris* is known to have microbiota consortia that interact and use nitrogen-based nutrients [27].

### 4.3. Co-Culture Analysis

Our co-culture experiments demonstrate clearly that both microorganisms, *C. vulgaris and Bacillus* spp., can coexist in a modified medium, without severe inhibition effects. While the growth rate of the two organisms together was lower than that of *C. vulgaris* grown separately, the joint cultures remained stable and did not collapse (Figure 5). Flow cytometry analysis yielded interesting results, revealing that, throughout the 16-day experiment period, the size of the bacterial group exceeded that of the algae. Initially, the bacterial population constituted 74.1%, whereas the algal group constituted 24.1% of the total population. Over the following two weeks, the bacterial component showed significant growth. The bacterial population was separated into two types, vegetative and spores; there were similar quantities at the beginning of the experiment (40.2%, 39.9%, respectively); however, after 16 days, spores were the majority. This clearly indicates stress conditions that should be taken into consideration for long-term co-culture experiments involving algae and bacteria.

## 5. Conclusions

We conducted experiments to assess the impact of hydrazine on the growth of algae and bacteria and analyzed its reduction in cultures. Both *C. vulgaris* and *Bacillus* spp., tested separately, successfully tolerated hydrazine at varying ppm levels. *C. vulgaris* can grow with up to 20 ppm hydrazine, and extremophilic bacteria from the Dead Sea region can accommodate 25 ppm. The mechanism through which algae can tolerate hydrazine is probably linked to their photosynthesis capability, while the mechanism for *Bacillus* spp. is unknown. Further studies are required to ascertain the exact nature of the photosystem in *C. vulgaris* and the effect of hydrazine on it. Mini plates of titanium did not have an adverse effect on growth or on hydrazine reduction.

We tested the co-culture of *C. vulgaris* and *Bacillus* spp. (ISO-36, closely related to *B. cereus*) and found that they can coexist in a modified medium (Bristol + LB) without harsh or negative toxic effects. This research demonstrates the potential for reusing hydrazine-based fuel tanks, creating a new type of photobioreactor integrated into life support systems on distant colonies.

Future work may include co-culture with hydrazine, and metagenomic and transcriptome analyses to identify potential bacterial metabolic pathways in coping with hydrazine. Chemical analysis of the main products from the hydrazine degradation process can also contribute to our understanding of the metabolic pathway. Furthermore, conducting extensive experiments in a 45 L bioreactor will add to the assessment of the potential for bioremediation with both microorganisms.

## Figures and Tables

**Figure 1 life-15-01197-f001:**
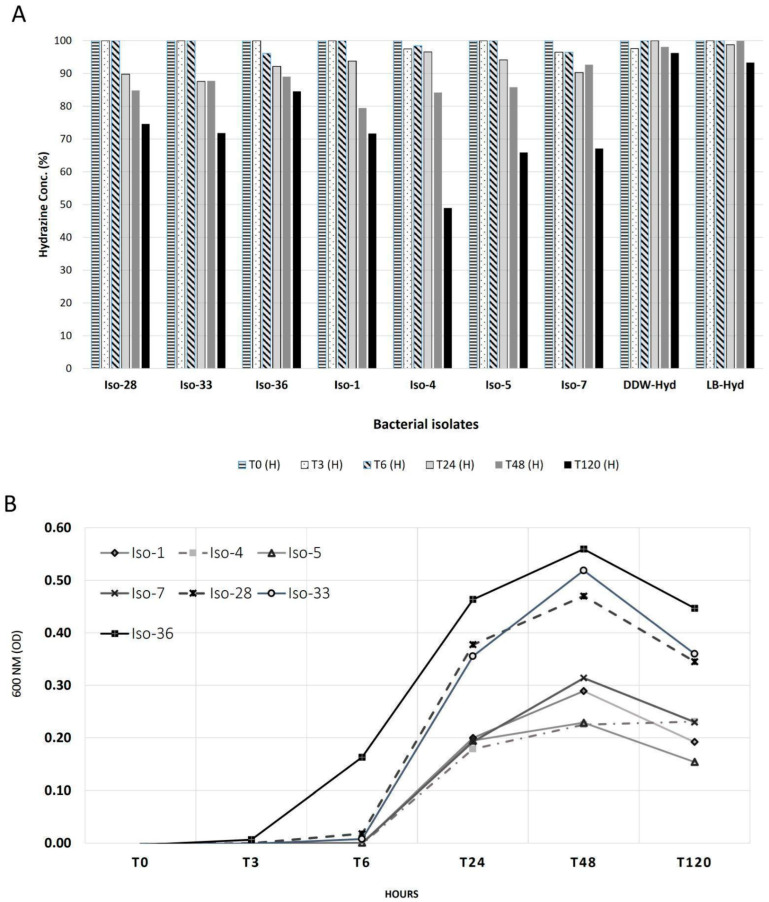
Bacterial hydrazine growth results: (**A**) Hydrazine concentration changes during 120 h with ISO-1, 4, 5, 7, 28, 33, and 36 compared to controls: DDW/LB-Hyd (25 ppm *v*/*v* in DDW/LB). (**B**) Growth curves of all isolates during the hydrazine consumption experiment. Measured at 600 nm.

**Figure 2 life-15-01197-f002:**
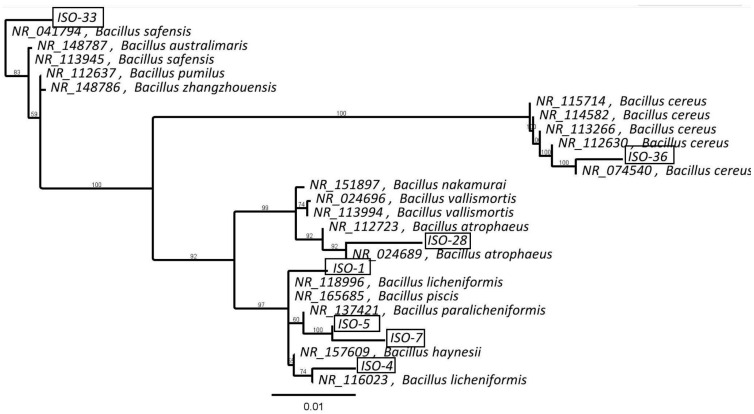
Phylogenetic analysis of bacterial isolates. An unrooted phylogenetic tree based on amplified 16S rRNA gene sequences is shown for isolates ISO-1, ISO-4, ISO-5, ISO-7, ISO-28, ISO-33, and ISO-36. The tree is based on consensus sequences derived from triplicates using a majority greedy clustering method. The phylogeny was inferred using the Tamura–Nei distance model and the neighbor-joining algorithm. Bootstrap values (from 100 replicates) are displayed at the nodes, representing branch support. The scale bar indicates 1% sequence divergence (0.01).

**Figure 3 life-15-01197-f003:**
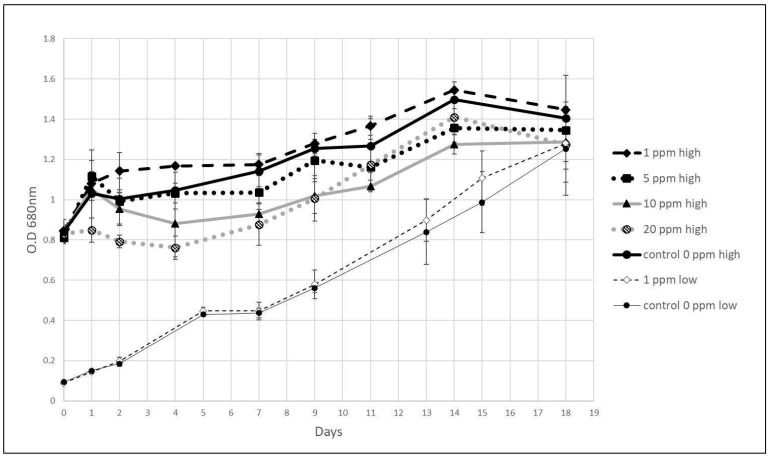
*C. vulgaris* growth at two initial culture densities and various hydrazine levels: 0.1 (=low) and 0.8 (=high) at OD 680 nm, hydrazine concentrations 1, 5, 10, and 20 ppm. Control w/o hydrazine. (*n* = 3; error bars are STDEV).

**Figure 4 life-15-01197-f004:**
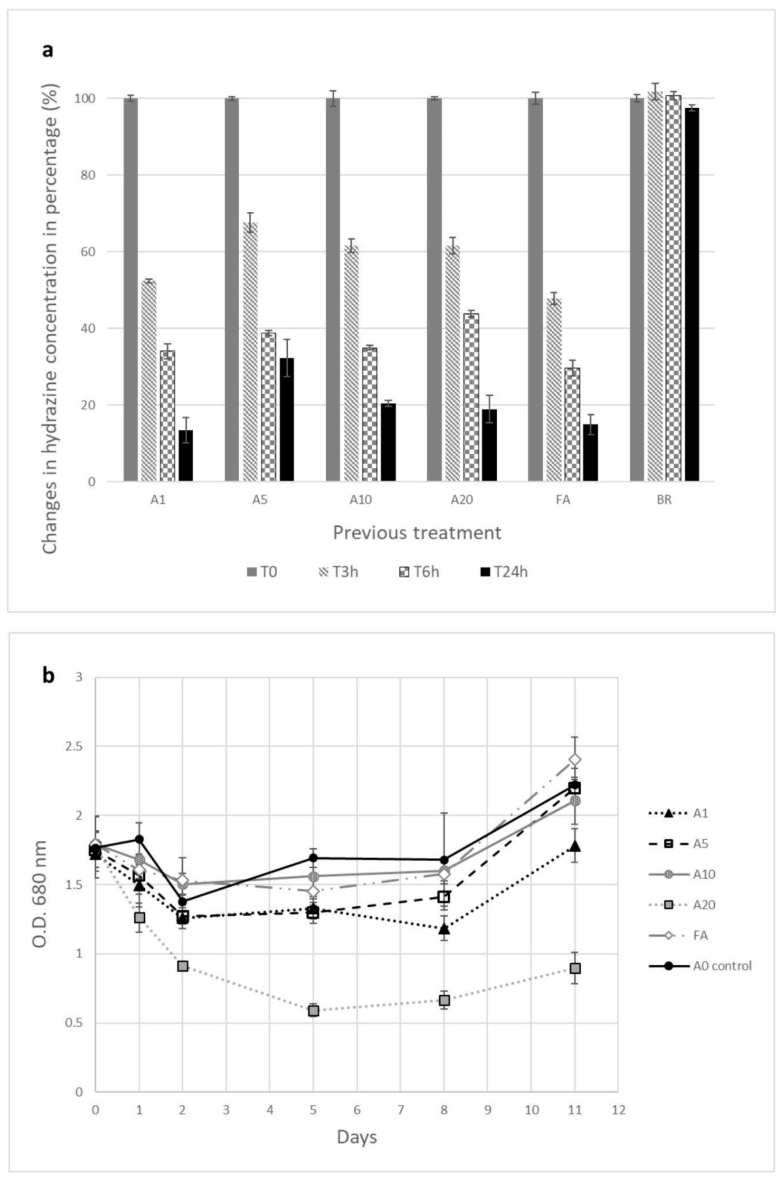
HT-2 results. A1, 5, 10, 20—algal cultures reintroduced to hydrazine concentration of 20 ppm after pre-exposure to 1, 5, 10, and 20 ppm. FA—fresh algae culture with no pre-exposure. Two controls: A0—algal culture w/o hydrazine, BR- Bristol medium with hydrazine 20 ppm w/o algae. (**a**) Hydrazine concentration (%) in algal cultures measured at 0, 3, 6, and 24 h. (**b**) *C. vulgaris* growth graph over 11 days. All cultures were grown at 20 ppm (experiments as well as measurements were performed in duplicate; error bars are STDEV).

**Figure 5 life-15-01197-f005:**
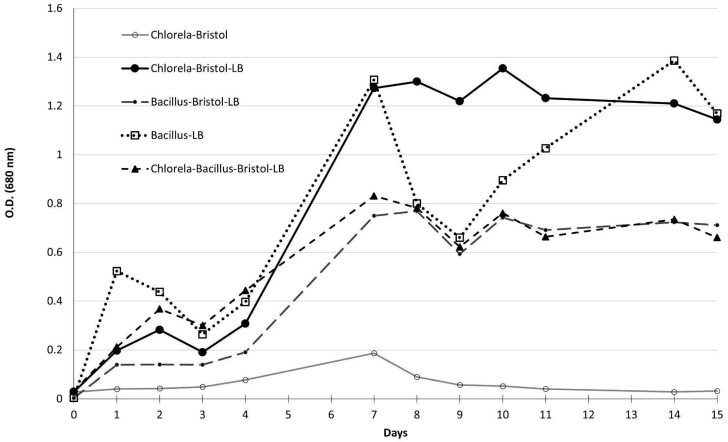
*C. vulgaris* and ISO-36 bacterial growth profiles without hydrazine: in different media—LB, Bristol x2, LB-Bristol x2 (20/80) over 15 days.

## Data Availability

Not applicable. All data used in this MS was supplied in text, figures and Appendix A.

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
