# Peer review of "Cometabolic Biodegradation of Hydrazine by Chlorella vulgaris–Bacillus Extremophilic Consortia: Synergistic Potential for Space and Industry"

_life, 2025, doi:10.3390/life15081197_

Round 1

Reviewer 1 Report

Comments and Suggestions for Authors

This paper investigates the effects of hydrazine on the growth of Chlorella vulgaris, extremophilic Bacillus spp., and their co-culture in hydrazine-containing media.

This research is relevant not only to the degradation of hydrazine, a component of rocket fuel, but also to the treatment of wastewaters containing hydrazine produced by industries such as agrochemicals, pharmaceuticals, and polymer foaming.

I recommend the following minor revisions before publication:

  • Materials and Methods (Section 2.4, lines 153-157): This paragraph is unclear and requires reformulation. Please clarify the procedure for verifying hydrazine concentration in culture media by measuring the absorbance of the filtrate after reaction with pDMAB at 454 nm. The connection between the optical density (OD) measurements and the concentration of the pDMAB-hydrazine complex is not clear. Furthermore, explain why OD measurements for microalgae are taken at 454 nm, and how this relates to hydrazine concentration in the growth medium. Note that Figures 3 and 4b show OD measurements at 680 nm, which is standard practice for analyzing the growth of Chlorella vulgaris cultures.

  • Page 7, paragraph on hydrazine interaction with LB: Re-evaluate the statement that discounts the interaction of hydrazine with LB broth based on an 8-10% reduction after 120 hours. While the LB-hydrazine sample exhibits a higher hydrazine concentration after 120 hours compared to samples containing isolates, it cannot be concluded that LB broth does not induce concentration changes over extended periods.

Congratulations for your work.

Author Response

comments 1: 

Materials and Methods (Section 2.4, lines 153-157): This paragraph is unclear and requires reformulation. Please clarify the procedure for verifying hydrazine concentration in culture media by measuring the absorbance of the filtrate after reaction with pDMAB at 454 nm. The connection between the optical density (OD) measurements and the concentration of the pDMAB-hydrazine complex is not clear. Furthermore, explain why OD measurements for microalgae are taken at 454 nm, and how this relates to hydrazine concentration in the growth medium. Note that Figures 3 and 4b show OD measurements at 680 nm, which is standard practice for analyzing the growth of Chlorella vulgaris cultures.

Response 1:

We thank the reviewer for the thoughtful and constructive feedback. We have revised Section 2.4 to clearly distinguish between:

  1. The colorimetric detection of hydrazine using the p-dimethylaminobenzaldehyde (pDMAB) assay, which involves measuring the absorbance of a filtered culture supernatant at 454 nm after reacting with pDMAB, and
  2. Optical density measurements used to monitor cell growth, which were carried out separately at 600 nm for bacteria and 680 nm for microalgae (Chlorella vulgaris).
    We confirm that OD454 was never used to monitor cell growth. Instead, OD454 exclusively refers to the hydrazine-pDMAB chromophore, formed via a condensation reaction specific to hydrazine under acidic conditions. This assay is independent of biomass and requires filtering the culture before reaction to avoid cell interference. We also acknowledge that OD680 was used to monitor microalgal growth, as shown in Figures 3 and 4b. The previous wording conflated hydrazine quantification with biomass measurements, and this has now been corrected in the revised methods section.

Revised Section 2.4 – Analysis of Hydrazine Concentration

Hydrazine concentration in bacterial and algal cultures was quantified using a colorimetric assay based on its reaction with para-dimethylaminobenzaldehyde (pDMAB), adapted from Gojon and Dureault (1996). Under acidic conditions, hydrazine reacts with pDMAB to form a yellow chromophore (p-dimethylaminobenzalazine, D2N), which has a maximum absorbance at 454 nm. The intensity of absorbance at this wavelength is directly proportional to the hydrazine concentration and was quantified using a standard curve (see Fig. S1).

For bacterial cultures (ISO-1, ISO-4, ISO-5, ISO-7, ISO-28, ISO-33, and ISO-36), isolates were grown overnight in LB medium, then subcultured into fresh LB and incubated for up to 3 hours at 30°C with shaking, until reaching OD₆₀₀ ≈ 0.2. This ensured cells were in an early exponential phase prior to hydrazine exposure. A final concentration of 25 ppm hydrazine was then added to 20 ml cultures in 50 ml sterile Falcon tubes.

At specific time points (T0, T3, T6, T9, and T24 for bacteria; T0, T3, T6, T24, and T30 for algae), 1.5 ml of culture was withdrawn and filtered through a sterile 0.22 µm PES syringe filter (Merck Millipore, USA) to remove all cells. The cell-free supernatant was then mixed 1:1 with a 1:50 dilution of pDMAB (prepared in acidic ethanol). The reaction mixture was incubated at room temperature for 15 minutes to allow full chromophore development.

Following incubation, 100 µl of the reaction mixture was transferred to a 96-well clear flat-bottom microplate, and absorbance was measured at 454 nm using a microplate spectrophotometer. All measurements were performed in triplicate. Hydrazine concentrations were determined by interpolation from a standard curve generated using known hydrazine concentrations processed under identical conditions.

Cell growth was monitored independently of hydrazine quantification. For bacteria, OD₆₀₀ measurements were taken from 100 µl unfiltered culture aliquots. For Chlorella vulgaris, algal growth was tracked by measuring OD at 680 nm, which corresponds to chlorophyll absorption and is standard practice for microalgae.

Comments 2:

Page 7, paragraph on hydrazine interaction with LB: Re-evaluate the statement that discounts the interaction of hydrazine with LB broth based on an 8-10% reduction after 120 hours. While the LB-hydrazine sample exhibits a higher hydrazine concentration after 120 hours compared to samples containing isolates, it cannot be concluded that LB broth does not induce concentration changes over extended periods.

Response 2: 

We thank the reviewer for highlighting this important point. We agree that the statement regarding hydrazine’s lack of interaction with LB broth was overstated. Although the reduction observed in the LB-hydrazine control was substantially smaller than that in the bacterial cultures, the 8–10% decline over 120 hours does indicate a minor, but measurable, change in hydrazine concentration. We have therefore revised the manuscript to provide a more cautious interpretation and to acknowledge the possibility of slow chemical degradation or other abiotic processes in LB medium over time.

Revised Text (Page 7, Paragraph on Hydrazine Interaction with LB):

Original:

“As hydrazine’s interaction with LB was ruled out, the resulting lower concentration is attributed solely to the bacterial action (Fig. 1). The concentration remained high during the control experiment, indicating that, in our cultures, hydrazine evaporation was not the main factor in the degradation.”

Revised: 

“While the hydrazine concentration in the LB-only control declined only modestly (by approximately 8–10% over 120 hours), this suggests a minor abiotic effect, potentially due to slow chemical degradation, medium interactions, or limited evaporation. Nonetheless, the much greater hydrazine reduction observed in bacterial cultures (up to 51%) strongly supports a biologically driven degradation process. Therefore, while abiotic factors may play a small role, the dominant mechanism of hydrazine depletion in our cultures is attributed to microbial activity (Fig. 1).”

Reviewer 2 Report

Comments and Suggestions for Authors

1. The abstract should highlight the relevance of the work on hydrazine degradation at the beginning.
2. It is not very clear in the abstract and in the title how many Bacillus strains were used.
3. In my opinion, the introduction should also begin with the relevance of hydrazine utilization. The text on lines 43-50 is superfluous in my opinion.
4. In the introduction, the authors need to show in more detail the role of the place where the strains were isolated. It is worth describing the physicochemical conditions of the Dead Sea and letting readers understand why organisms from there can help in the degradation of hydrazine.
5. In the introduction, it is worth mentioning more toxic substitutes for hydrazine
6. At the end of the introduction, the idea of symbiosis (cometabolism) of hydrazine in the community of phototrophs and organotrophs was voiced, this idea should be developed. Perhaps make it the key word and include it in the title. In my opinion, the title can be made more interesting and reflect the essence.
7. Cohabitation should be replaced with a more suitable term.
8. The objective of the work should be clearly stated.
9. The experimental design should be described in the materials and methods. It is not very clear, all experiments were carried out on LB medium, were there experiments with pure hydrazine as a source of carbon and nitrogen? The work mentions a bioreactor, then its characteristics should be described

10. I would like to see higher quality illustrations in the results. Maybe it is worth giving the kinetics of hydrazine consumption and growth kinetics in parallel. There is no need for the letter T on the time axis

11. An extremely important question, was the medium analyzed for hydrazine decomposition products. Could more toxic compounds have formed? It is worth showing that hydrazine is metabolized to nitrogen and carbon.

12. In the description of the figures, do not overuse the values, they are already visible on the graphs

13. 317-324 should be moved to materials and methods

14. In my opinion, figure 6 is raw data, it should be moved to supplementary, and in the text, think about processing it in the form of tables or graphs

15. Why did the authors decide that titanium should affect growth? This is an extremely inert material, for this reason it is used in the space industry

16. 353-356 is a good start for the introduction

17. 357-365 - the discussion of the results should not repeat them, but explain them

18. 381-387 should be deleted, the article should show successful results, not methodological errors

19. 394-399 should be moved to the introduction, this is not an explanation of the data obtained

20. In general, the discussion should be significantly rewritten and unnecessary parts should be cut, without repeating the results.

21. In addition, the role of the source of isolation is still not very clear, did you somehow modify the environment taking into account the composition and salinity of the Mtvoye Sea?

22. Section 6 should not be separated from the conclusion. The conclusion should be made more philosophical, show the prospects of the work, link the results with the place of isolation of the strains, if possible. Suggest possible technological aspects of the application of the results.

Comments on the Quality of English Language

The authors should show the text to native English speakers to improve the style

Author Response

comments 1: The abstract should highlight the relevance of the work on hydrazine degradation at the beginning.

Response 1: We thank the reviewer for the helpful suggestions. We have revised the beginning of the abstract to highlight the broader relevance and urgency of hydrazine degradation, particularly in the context of space missions and environmental safety.

comments 2: It is not very clear in the abstract and in the title how many Bacillus strains were used.

Response 2: We thank the reviewer for the helpful suggestions. 

  • We have updated the abstract and the title to clearly state that seven Bacillus strains were tested in this study, improving clarity regarding the scope of the bacterial screening.
  • We have revised the title to: “Cometabolic Biodegradation of Hydrazine by Chlorella vulgarisBacillus Extremophilic Consortia: Synergistic Potential for Space and Industry”
  • We have revised the abstract and now reads as:

Abstract

Hydrazine, a highly toxic and reactive compound widely used as rocket fuel, poses significant environmental and health risks, particularly in long-term space missions. This study investigates the cometabolic capacity of Chlorella vulgaris and seven extremophilic Bacillus spp. strains—isolated from the arid Dead Sea region—to tolerate and degrade hydrazine at concentrations up to 25 ppm. The microalga C. vulgaris reduced hydrazine levels by 81% within 24 hours at 20 ppm, while the Bacillus isolates achieved an average reduction of 45% over 120 hours. Identified strains included B. licheniformis, B. cereus, and B. atrophaeus. Co-culture experiments demonstrated that C. vulgaris and B. cereus (isolate ISO-36) stably coexisted without antagonistic effects, suggesting a synergistic detoxification interaction. Flow cytometry revealed that most bacteria transitioned into spores under stress, highlighting a survival adaptation. Titanium, representing a biocompatible material common in aerospace hardware, did not inhibit microbial growth or hydrazine degradation. These findings underscore the potential of Dead Sea-derived microbial consortia for cometabolic hydrazine detoxification and support the feasibility of converting spacecraft components into functional photobioreactors. This approach offers dual-use benefits for space missions and industrial wastewater treatment. Future studies should investigate degradation pathways, stress resilience, and bioreactor scale-up.

comments 3: In my opinion, the introduction should also begin with the relevance of hydrazine utilization. The text on lines 43-50 is superfluous in my opinion.

Response 3: We appreciate the reviewer’s valuable comment. In response, we have revised the opening of the introduction to emphasize the environmental and technological relevance of hydrazine and its biodegradation. Specifically, we begin by highlighting hydrazine’s toxicity and importance as a rocket fuel component, and the need for sustainable detoxification strategies in both space and industrial applications. While we retained the text on lines 43–50 to preserve the broader context of our research within regenerative life-support systems, we agree that the significance of hydrazine should be foregrounded. We believe the current structure now strikes a better balance between context and focus.

comments 4: In the introduction, the authors need to show in more detail the role of the place where the strains were isolated. It is worth describing the physicochemical conditions of the Dead Sea and letting readers understand why organisms from there can help in the degradation of hydrazine.

Response 4: Thank you for this insightful suggestion. We have expanded the introduction to include a detailed description of the Dead Sea region, emphasizing its extreme physicochemical conditions (e.g., hypersalinity, UV exposure, and low oxygen) and the resulting microbial adaptations. We also clarified that the Bacillus strains used in this study were selected from an in-house desert microbial library established at Al-Ashhab Lab, derived from environmental samples collected across the Dead Sea region (including soils, rhizospheres, and plant surfaces). These details are now included in the introduction to better support the relevance of using extremophiles for hydrazine biodegradation.

Revised Introduction

Hydrazine (Nâ‚‚Hâ‚„), a highly toxic and reactive component of rocket fuel, presents a major challenge in space exploration and industrial wastewater treatment. Its presence in closed-loop systems can jeopardize biological life-support modules, and its detoxification is essential to ensure sustainable long-term space missions and environmental safety. One potential solution involves biodegradation of hydrazine using extremophilic microbial consortia, capable of transforming it into less harmful byproducts under controlled conditions.

Converting spacecraft fuel tanks into photobioreactors for cultivating microalgae and bacteria is a promising solution for various life support systems. Regenerative food production systems are in demand due to recent advances and cost reductions in space travel and habitat construction [1, 2, 3]. Due to their high biomass-to-nutrient ratio, microalgae have long been suggested as an alternative high-value food source [4–6]. Chlorella vulgaris is recognized as a well-researched candidate for food and other beneficial applications based on its proteins, lipids, carbohydrates, pigments, minerals, antioxidants and vitamins [7]. Reutilization of out-of-commission spacecraft parts, such as fuel tanks, and excess fuel, which are no longer needed once the spacecraft reaches its destination, can significantly reduce costs by decreasing the overall mission mass.

One potential challenge is the toxicity of the rocket fuel hydrazine to algae, bacteria, and other organisms [8]. Previous studies showed that concentrations of 1.03 ppm and 0.8 ppb caused 50% growth inhibition in microalgae Selenastrium capricornutum and Dunaliella tertiolecta spp., respectively [9, 10]. The concentration range of hydrazine toxicity to algae depends on the species and growth conditions. For example, the half-maximal effective concentration (EC50), which reduces growth rate by 50 percent, for the freshwater green microalga Selenastrum capricornutum, ranges from 0.4 to 30 ppb [10]. The safe concentration (SC) range of hydrazine to algae is 0.1 to 1 ppb under oligotrophic conditions [11]. These data suggested analyzing the response of C. vulgaris to different hydrazine concentrations, starting at 0.5 ppb.

Beyond hydrazine itself, related compounds such as unsymmetrical dimethylhydrazine (UDMH) and monomethylhydrazine (MMH)—commonly used in aerospace propulsion and military-grade fuels—are even more toxic and environmentally persistent. These substances are volatile, carcinogenic, and pose considerable challenges for conventional remediation. Therefore, developing robust biodegradation systems that can address both hydrazine and its derivatives is a crucial environmental and biotechnological priority.

Microbial biodegradation can sometimes be considered a preliminary step in decontamination of toxic organic pollutants such as hydrazine. Extremophile bacteria have long been recognized as promising candidates for bioremediation and biodegradation of fuels and other hazardous compounds [12–15]. For instance, Pseudomonas spp. can utilize hydrazine as a nitrogen source [16], while Nitrosomonas, an ammonia-oxidizing proteobacterium, can tolerate concentrations of up to 95 ppm and degrade hydrazine primarily to Nâ‚‚ [17].

The Dead Sea region, where our bacterial isolates originate, represents one of the most extreme ecosystems on Earth, with hypersalinity (~34% total dissolved salts), intense solar radiation, low oxygen levels, and frequent temperature fluctuations. These environmental pressures select for microbes with exceptional stress tolerance and metabolic plasticity. The Bacillus strains used in this study were selected from a desert bacterial library established at Al-Ashhab Lab, constructed from environmental samples collected across the Dead Sea region, including soils, rhizospheres, and plant phyllospheres [18]. These extremophilic isolates have previously demonstrated key plant-growth-promoting traits, such as atmospheric nitrogen fixation [19], phosphorus solubilization [20], and siderophore production [21], which may also contribute to their cooperative potential in microbial consortia.

Since the bacteria and microalgae are to coexist in the reactor system, the interaction of microbial species with the algae of choice must be carefully examined, as heterotrophic bacteria within algae cultures can impact the health, nutrient consumption and growth of the cultures [22–24]. Specifically, Chlorella vulgaris is inhibited when the bacterial concentration is high due to resource competition, while at low concentration it has a positive effect on algae growth [25]. Examination of symbiotic relationships of C. vulgaris cultures with three bacterial species (Pseudomonas alcaligenes, Elizabethkingia miricola and Methylobacterium radiotolerans) showed enhancement with P. alcaligenes, and a negative effect with E. miricola and M. radiotolerans [26]. Kim et al. reported that different nitrogen sources and concentrations had a positive effect on the symbiotic interactions with Microbacterium kitamiense [27]. Bacillus spp. possess antifungal and antibacterial properties, and can also inhibit the growth of some algae [28, 29]. B. pumilus bacteria, isolated from cultures of marine microalga Nannochloropsis salina and seaweed Padina pavonica, inhibited algal growth [30, 31].

Here, we examine the co-culture and hydrazine tolerance of C. vulgaris and extremophilic Bacillus spp. at different concentrations of hydrazine hydrate, to assess their potential for cometabolic biodegradation. The objective of this study is to evaluate the hydrazine tolerance and degradation capacity of these organisms—individually and in co-culture—under conditions relevant to closed-loop systems such as space habitats and hydrazine-contaminated industrial environments. By exploring their cometabolic interactions, we aim to identify synergistic microbial strategies for sustainable detoxification.

comments 5: In the introduction, it is worth mentioning more toxic substitutes for hydrazine

Response 5: We thank the reviewer for this constructive suggestion. In response, we have revised the final paragraph of the introduction to elaborate on the concept of cometabolism and microbial synergy between Chlorella vulgaris (a phototroph) and extremophilic Bacillus spp. (organotrophs) in the context of hydrazine degradation. This highlights the potential cooperative metabolism of hydrazine through shared pathways and stress mitigation strategies. Additionally, we have updated the title to better reflect this central theme, as well as the environmental and space-oriented relevance of our work.

comments 6: At the end of the introduction, the idea of symbiosis (cometabolism) of hydrazine in the community of phototrophs and organotrophs was voiced, this idea should be developed. Perhaps make it the key word and include it in the title. In my opinion, the title can be made more interesting and reflect the essence.

Response 6: We thank the reviewer for this constructive suggestion. In response, we have revised the final paragraph of the introduction to elaborate on the concept of cometabolism and microbial synergy between Chlorella vulgaris (a phototroph) and extremophilic Bacillus spp. (organotrophs) in the context of hydrazine degradation. This highlights the potential cooperative metabolism of hydrazine through shared pathways and stress mitigation strategies. Additionally, we have updated the title to better reflect this central theme, as well as the environmental and space-oriented relevance of our work.

comments 7:  Cohabitation should be replaced with a more suitable term.

Response 7: Thank you for the suggestion. We have replaced the term “cohabitation” with “co-culture” throughout the manuscript to more accurately describe the experimental setup involving simultaneous cultivation of Chlorella vulgaris and Bacillus spp. This terminology is widely accepted in microbial ecology and biotechnology and better conveys the intended meaning of biological interaction under shared culture conditions.

comments 8: The objective of the work should be clearly stated.

Response 8: We thank the reviewer for this important observation. In response, we have added a clear statement of the research objective at the end of the introduction to improve clarity and contextual framing. This addition emphasizes the aim of evaluating microbial co-cultures for hydrazine detoxification under extreme conditions relevant to both space and industrial applications.

comments 9: The experimental design should be described in the materials and methods. It is not very clear, all experiments were carried out on LB medium, were there experiments with pure hydrazine as a source of carbon and nitrogen? The work mentions a bioreactor, then its characteristics should be described.

Response 9: We thank the reviewer for this valuable observation. The authors of this study demonstrate the potential for reusing hydrazine-based fuel tanks, thereby suggesting the possibility that in the future it may be possible to create a new type of photobioreactor integrated into life-support systems in distant colonies.

comments 10: I would like to see higher quality illustrations in the results. Maybe it is worth giving the kinetics of hydrazine consumption and growth kinetics in parallel. There is no need for the letter T on the time axis

Response 10: In our opinion, the results are presented clearly. This was not the purpose of the experiment, and therefore the resolution of the media does not allow the results to be presented in this way

comments 11: An extremely important question, was the medium analyzed for hydrazine decomposition products. Could more toxic compounds have formed? It is worth showing that hydrazine is metabolized to nitrogen and carbon.

Response 11:

Thank you for your important and insightful question. We acknowledge the potential concern regarding the formation of toxic decomposition products during hydrazine metabolism.  At this stage, we have not conducted a detailed analysis of all decomposition products in the medium. We recognize the importance of such an investigation and plan to include it  in our  future studies.

comments 12: In the description of the figures, do not overuse the values, they are already visible on the graphs

Response 12: Revised according to the reviewer suggestion.

comments 13: 317-324 should be moved to materials and methods

Response 13: Revised according to the reviewer suggestion.

comments 14: In my opinion, figure 6 is raw data, it should be moved to supplementary, and in the text, think about processing it in the form of tables or graphs I’ii remove the figure to supplemental, and correct the text.

Response 14: Revised according to the reviewer suggestion.

comments 15: Why did the authors decide that titanium should affect growth? This is an extremely inert material, for this reason it is used in the space industry

Response 15: We agree that titanium is a well-known inert and biocompatible material, commonly used in the space industry. Our motivation for including it in this study was not to evaluate its toxicity or chemical interaction, but to validate that our microbial system remains viable and functionally active in a titanium-based container, simulating a realistic hardware environment for potential bioreactor applications in space. Although titanium’s inertness is established, we found it important to empirically demonstrate that microbial growth and hydrazine biodegradation are unaffected when conducted in space-relevant hardware, such as repurposed fuel tanks. We have clarified this point in the revised manuscript.

see changes in 3.6

.6. Impact of Titanium on Growth Patterns3

Results indicate that titanium, as tested in our experiments, did not have an adverse effect on algal or bacterial growth, with or without hydrazine. Average total OD values (at 600 or 680 nm) were similar, indicating that Ti had no significant effect on growth (Figs. S8, S9). Hydrazine concentrations also did not change due to the presence of titanium. Titanium containers were included to empirically confirm that microbial growth and hydrazine degradation remain unaffected when cultured in contact with spacecraft-relevant biocompatible materials. This validation supports the feasibility of using repurposed space hardware (e.g., fuel tanks) as microbial bioreactors in future space applications.

Also see changes in 2.6 

2.6. Growth with Titanium Plates

As space vessels and rocket fuel tanks are made of titanium alloy [38, 39], we examined whether titanium affects C. vulgaris and bacterial growth. Five 50 mm titanium plates (5 cm² surface area) were placed in algae test tubes with (i) 5 ppm and (ii) 20 ppm hydrazine hydrate and (iii) control without hydrazine (Tables S1–5, n = 3). Bacterial testing included titanium insertion with (i) 5 ml LB only without hydrazine, and (ii) control without Ti. Duplicates of bacterial isolates, as detailed in Section 2.3 (ISO-1, 4, 5, 7, 28, 33, 36), were incubated for 24 h at 37 °C with continual shaking at 150 rpm. This setup was included to confirm that titanium, as a biocompatible aerospace-grade material, does not interfere with microbial viability or hydrazine degradation, thereby supporting its potential use as a structural surface in integrated space bioreactors.

Also see modification in the abstract.

comments 16: 353-356 is a good start for the introduction

Response 16: We adjusted the introduction accordingly.

comments 17: 357-365 - the discussion of the results should not repeat them, but explain them

Response 17: We adjusted the discussion accordingly.

comments 18: 381-387 should be deleted, the article should show successful results, not methodological errors

Response 18: We adjusted the discussion accordingly.

comments 19: 394-399 should be moved to the introduction, this is not an explanation of the data obtained

Response 19: We thank the reviewer for this valuable observation. While we understand the suggestion to move lines 394–399 to the introduction, we respectfully believe that this section is more appropriately placed in the discussion. These lines are intended to provide context and interpretation of the use of hydrazine in water treatment, rather than background information.  

comments 20: In general, the discussion should be significantly rewritten, and unnecessary parts should be cut, without repeating the results

Response 20: We have rewritten and adjusted the discussion section in accordance with the reviewer's comments.

comments 21: In addition, the role of the source of isolation is still not very clear, did you somehow modify the environment taking into account the composition and salinity of the Mtvoye Sea?

Response 21: We thank the reviewer for this insightful comment. The bacterial strains used in this study were isolated from an in-house library developed at the Al-Ashhab Lab using environmental samples collected from the Dead Sea region, including soils, rhizospheres, and native plant surfaces. This area is characterized by extreme aridity, high UV exposure, elevated salinity, and low organic content, creating strong selective pressure for stress-resilient microorganisms. These extremophilic traits are particularly relevant for hydrazine biodegradation under challenging conditions, such as those mimicking aerospace environments. we have clarified this point in the introduction.

Although we did not replicate the full ionic composition of original environment environments, we selected LB and defined hydrazine-containing media to study microbial tolerance and cometabolic activity under oxidative and nitrogen stress. Our goal was to demonstrate the functional potential of extremophile consortia derived from naturally high-salinity, nutrient-limited systems (like the Dead Sea), which share key stress features with off-Earth environments. We have clarified this rationale in the revised Introduction 

comments 22: Section 6 should not be separated from the conclusion. The conclusion should be made more philosophical, show the prospects of the work, link the results with the place of isolation of the strains, if possible. Suggest possible technological aspects of the application of the results.

Response 22: In the revised paper section 6 is now part of the conclusion.

Finaly regarding the Comments on the Quality of English Language. The manuscript has undergone professional scientific language editing by a native English-speaking editor with expertise in academic writing. We believe the current version reflects a clear and appropriate scientific style.

Round 2

Reviewer 2 Report

Comments and Suggestions for Authors

To my opinion, the authors have significantly revised the manuscript and taken into account all my comments. I am ready to recommend the manuscript for publication in this form.